# Research on the Corrosion Inhibition Behavior and Mechanism of 1-Hydroxy-1,1-ethyledine Disodium Phosphonate under an Iron Bacteria System

**Ping Xu** *, **Yuxuan Zhao** †  and **Pengkai Bai** †

Key Laboratory of Urban Stormwater System and Water Environment, Ministry of Education, Beijing University of Civil Engineering and Architecture, Beijing 100044, China; 15930170249@163.com (Y.Z.); baipengkai0508@163.com (P.B.)
*  Correspondence: xuping@bucea.edu.cn
†  These authors contributed equally to this work.

**Abstract:** Regenerated water serves as a supplementary source for circulating cooling water systems, but it often fosters microbial growth within pipelines. Given its widespread use as a corrosion inhibitor, understanding HEDP's efficacy in microbial environments and its impact on microorganisms is imperative. This study established an iron bacterial system by isolating and enriching iron bacteria. Through a comprehensive approach incorporating corrosion weight loss analysis, XPS analysis, SEM electron microscopy, as well as microbial and electrochemical testing, the corrosion inhibition behavior and mechanism of HEDP within the iron bacterial system were investigated. The findings reveal that within the iron bacterial system, HEDP achieves a corrosion inhibition rate of 76% following four distinct stages—weakening, strengthening, stabilizing, and further strengthening—underscoring its robust corrosion inhibition capability. Moreover, HEDP enhances the density of biofilms and elevates the activation energy of carbon steel interfaces. It alternates with oxygen to continuously suppress the activity of IRB while gradually inhibiting the activity of IOB. This process culminates in a corrosion inhibition mechanism where cathodic inhibition predominates, supported by anodic inhibition as a complementary mechanism.

**Keywords:** regenerated water; iron bacterial system; HEDP; corrosion inhibition mechanism; corrosion inhibition process



## 1. Introduction

Pipeline corrosion poses a significant challenge in industrial cooling and circulating water systems, with the potential to cause severe production accidents when it affects critical areas [1,2]. In response to the growing global emphasis on water conservation, many enterprises are increasingly turning to recycled water as a supplementary source for their circulating cooling water systems. However, recycled water often exhibits certain characteristics, such as a diverse ion species and high organic matter concentration, that create a conducive environment for microbial growth and reproduction within the pipelines [3]. This heightened microbial activity elevates the risk of microbial corrosion (MIC) in circulating cooling water systems.

In combating corrosion, the addition of corrosion inhibitors to the system stands out as a common practice. HEDP, owing to its capability to chelate iron ions within corrosive environments, fosters the formation of a dense passivation film, thereby retarding pipeline substrate corrosion, and it is widely utilized in circulating cooling water systems [4]. Studies have revealed HEDP's notable synergistic effect, effectively interacting with various phosphate, polymer, and copolymer corrosion inhibitors [5]. Moreover, when paired with zinc ions, sodium tungstate, sodium silicate, and specific rare earth elements, the corrosion inhibition rate experiences significant enhancement [6–9]. Nonetheless, some researchers

have observed that the film generated by HEDP on the metal surface might contain irregular pores [10], potentially influencing the corrosion inhibition efficacy within microbial systems.

Currently, most research on the impact of HEDP on microorganisms has been concentrated in the medical field, analyzing its effects on inhibiting microbial activity, disrupting biofilm structure, and inhibiting mycelial growth at various concentrations and combinations [11–14]. However, only three published articles are related to corrosion. Liang et al. examined the adhesion of surface SRB on pipes treated with HEDP using different methods. Within 4 h of initial adhesion, they found that the presence of HEDP reduced the adhesion kinetics constant of SRB on the pipe surface [15]. Qi et al. investigated the effects of different chemical agents on the corrosion process of SRB, focusing on stages with higher SRB metabolic activity based on changes in polysaccharide content on carbon steel surfaces. At this stage, they discovered that HEDP can alter the content of proteins and polysaccharides in SRB biofilms, increase the corrosion voltage of stainless steel, and enhance its resistance to pitting corrosion [16]. Li et al. analyzed the effects of SRB on the corrosion behavior of pipes in the presence and coexistence of HEDP and D-Phe. After 10 days of corrosion, it was observed that HEDP slowed down the cathodic reaction rate and inhibited the expression of certain genes in the attached state of SRB [17].

The existing research primarily focuses on investigating the impact of HEDP on the corrosion behavior of SRB attached to pipe surfaces during specific corrosion stages. However, microbial growth is a dynamic process, necessitating research on each stage of the corrosion process to comprehend the varying effects of HEDP on microorganisms. Additionally, circulating cooling water systems harbor diverse bacteria, with potential synergistic or antagonistic effects that could modify HEDP's efficacy [18,19]. Because SRB represents a bacterial community with a relatively straightforward corrosion mode, enriching the types of experimental bacteria is crucial to encompass a broader spectrum of corrosion behaviors. Moreover, HEDP influences both surface and suspended microorganism growth, yet existing research lacks comprehensive analysis integrating the growth of suspended microorganisms.

This study employed iron bacteria, commonly found in circulating cooling water systems [20–23], as experimental strains due to their broader range of corrosion types compared to SRB [24,25]. The experimental duration was appropriately extended, with increased emphasis on detecting and analyzing suspended iron bacteria and changes in carbon steel anode reactions. The effects of HEDP on various parameters, including the growth rate of iron bacteria, the types and composition of corrosion products, the biofilm structure and composition, reaction kinetics between the cathode and anode, corrosion current density, and AC impedance, were analyzed using a corrosion weight loss method, electrochemical testing, XPS analysis, and SEM analysis across different stages of iron bacterial growth. These investigations comprehensively elucidated the corrosion inhibition characteristics and potential mechanisms of HEDP in iron bacterial systems.

## 2. Materials and Methods

### 2.1. Materials

The test specimens utilized for dynamic corrosion experiments were custom-made National Type I carbon steel specimens (GB/T 699-2015, [26]) boasting dimensions of 50 mm × 25 mm × 2 mm. These specimens were crafted from #20 carbon steel characterized by predominant constituents, such as 99.421% Fe, 0.095% C, 0.17% Si, 0.29% Mn, and 0.012% S. Prior to experimentation, a meticulous pre-treatment protocol was followed: immersion in acetone, wiping with degreased cotton, drying using anhydrous filter paper, rinsing with distilled water, submerging in anhydrous ethanol for 1 min, another wipe with degreased cotton, drying with $N_2$ for 3 min, wrapping with anhydrous filter paper, placement in a drying dish for 24 h, and subsequent weighing for readiness. Furthermore, prior to usage, the specimens underwent a 30 min disinfection process under ultraviolet light.

The working electrode employed for electrochemical testing was a 1 cm$^2$ self-fabricated carbon steel electrode encased within epoxy resin with a dedicated 1 cm$^2$ surface area designated for electrochemical assessments. Prior to utilization, meticulous preparatory steps were executed. Initially, the exposed electrode surface underwent polishing using 180#–2000# sandpaper, followed by a refinement process utilizing 0.05 mm of Al$_2$O$_3$ polishing powder and suede cloth until achieving a mirror-like finish. Subsequently, the electrode was thoroughly rinsed with distilled water, subjected to dehydration in anhydrous ethanol for 1 min, and then placed in a drying dish for 24 h before being deemed ready for use.

The experimental iron bacteria were sourced from the bacterial repository of Huazhong University of Science and Technology. These bacteria underwent cultivation in an environment emulating circulating cooling water maintained for a duration of one month. The water quality parameters are detailed in Table 1. Post-cultivation, the cooling water containing bacteria was extracted and subjected to preliminary separation using the MPN method (GB/T 14643.6-2009, [27]). Positive test tubes were then collected, and iron bacteria liquid culture medium was introduced, ensuring thorough mixing. Following a 14-day incubation period at 29 ± 1 °C, the culture medium displayed stratification, with a clear upper layer and iron hydroxide flocculent sediment settling at the bottom. After remixing, a portion of the culture medium was inoculated onto a solid iron bacteria culture medium, prepared with a 15‰ agar ratio, followed by another 14-day incubation at 29 ± 1 °C. Colonies exhibiting a metallic luster on the culture medium were selectively picked and further cultivated in iron bacteria liquid culture medium. This process was iterated 3–4 times to enrich relatively pure iron bacteria, which served as the experimental bacteria for the study.

**Table 1.** Simulation of circulating cooling water quality.

|  | CaCl$_2$ | Na$_2$SO$_4$ | NaHCO$_3$ | NaNO$_3$ | K$_2$HPO$_4$ | (NH$_4$)$_2$SO$_4$ | pH |
|---|---|---|---|---|---|---|---|
| Concentration (mg/L) | 693.7 | 340.2 | 619.6 | 60.0 | 7.0 | 10.0 | 7.9 |

### 2.2. Experimental Devices

The corrosion simulation experiment of carbon steel was conducted within a custom-built rotary hanging plate reactor, illustrated in Figure 1. The reactor was equipped with a stirring device, simulating the hydraulic flow within actual pipeline networks, along with a temperature control system. Additionally, a water pump regulated the inlet water, maintaining a volume of 2.5 L. The device operated at a temperature of 30 °C and a speed of 60 r/min to replicate the hydraulic conditions of circulating cooling water pipe networks. To ensure stability in both water volume and quality, simulated circulating cooling water was replenished into the device every 2 h, as outlined in Table 1.

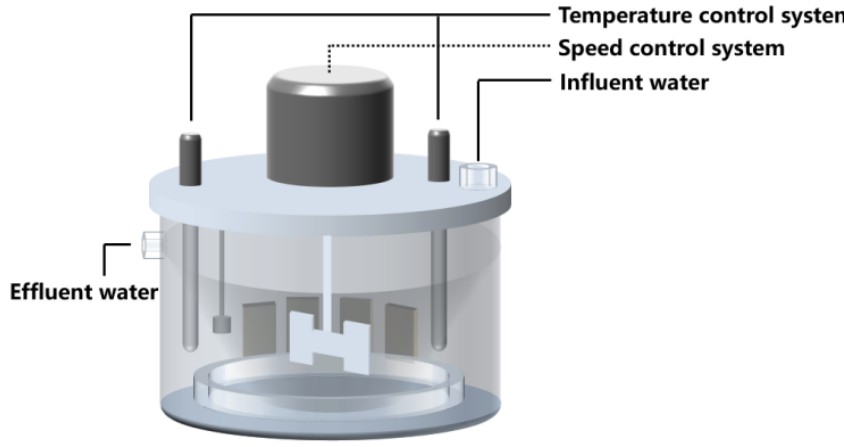

**Figure 1.** Artificial reactor.

Two sets of operating conditions were established, each spanning a duration of 15 days: (1) IB condition: simulated circulating cooling water was supplemented with $3.2 \times 10^5$ cfu/mL iron bacteria, along with the addition of carbon steel specimens and electrodes; (2) IB+HEDP condition: simulated circulating cooling water, $3.2 \times 10^5$ cfu/mL iron bacteria, and 30 mg/L HEDP were combined, followed by the inclusion of carbon steel specimens and electrodes.

To ensure the reliability of experimental outcomes, three parallel experiments were conducted within each group.

Throughout the experiment, assessments were conducted on the 1st, 3rd, 5th, 7th, 9th, 11th, and 15th days. These evaluations encompassed the quantification of suspended and attached iron bacteria, the determination of total extracellular polymeric substances (EPS) on the test piece surfaces, and analysis of protein and polysaccharide content within EPS components. Additionally, corrosion weight loss measurements were recorded at 1 d, 3 d, 7 d, 11 d, and 15 d intervals. Employing a traditional three-electrode system, polarization curves and electrochemical impedance spectroscopy (EIS) were generated for the electrodes. Subsequent examination of corrosion products on the surfaces of specimens at 3 d, 7 d, 11 d, and 15 d intervals was conducted via XPS and SEM analysis for comprehensive observation and analysis.

### 2.3. Detection Methods

The corrosion rate of carbon steel and the corrosion inhibition rate of HEDP were determined utilizing the weight loss method, employing the following formulas:

$$V = \frac{8760 \times 10 \times (m_1 - m_2)}{s \cdot t \cdot \rho} \tag{1}$$

$$\eta = \frac{V_1 - V_2}{V_1} \times 100\% \tag{2}$$

In the formula, $V$ denotes the corrosion rate (mm/a), while $m_1$ and $m_2$ signify the mass of the test piece before and after the experiment (g), respectively. $\rho$ represents the density of carbon steel (7.86 g/cm$^3$), $s$ denotes the surface area of the specimen (cm$^2$), and $t$ signifies the experimental time (h). $\eta$ denotes the corrosion inhibition rate (%), with $V_1$ and $V_2$ representing the corrosion rates of carbon steel at the same experimental time.

The number of suspended and attached iron bacteria was determined using the Most Probable Number (MPN) technique. This involved selecting three points from different areas within the reactor to collect water samples. These samples were thoroughly mixed, diluted according to gradients, and inoculated into an iron bacteria culture medium. Subsequently, they were incubated at $29 \pm 1$ °C for 48 h, after which the suspended iron bacteria were counted.

Concurrently, while collecting water samples, the test piece was retrieved, and the biofilm and corrosion products on its surface were carefully scraped off using a scraper. These samples were then placed in sterile sampling tubes. The process was repeated to count the number of attached iron bacteria, with the result indicating the quantity of attached iron bacteria.

To extract biofilm EPS, freeze-drying was employed. Firstly, corrosion products with biofilm on the test piece were scraped off using a scraper and placed in a centrifuge tube containing physiological saline. The supernatant was then subjected to centrifugation at 12,000 rpm for 20 min at 4 °C and filtered three times using a 0.22 μm filter membrane. The filtrate was subsequently frozen at $-50$ °C and then dried at the same temperature before being weighed.

For protein measurement in EPS, bovine serum albumin served as the standard, and Coomassie Brilliant Blue was utilized as the colorimetric solution. The protein content was determined via spectrophotometry at a wavelength of 595 nm. Polysaccharide quantifica-

tion was conducted using glucose as the standard by employing the phenol sulfuric acid method and measuring spectrophotometrically at a wavelength of 490 nm.

Electrochemical testing was performed using a CHI660C electrochemical workstation (Shanghai, China). A carbon steel electrode served as the working electrode, a platinum electrode as the auxiliary electrode, and a saturated calomel electrode as the reference electrode. For the polarization curve, the scanning range was set at open circuit voltage $\pm$ 0.3 V, with a scanning speed of 0.001 V/s. For the EIS analysis, the initial potential was maintained at 0 V, with a frequency range spanning from $10^5$ to $10^{-2}$ Hz and a disturbance potential of 5 mV.

The surface morphology of the specimen was examined using environmental scanning electron microscopy (SEM) (QUANTA 200F, The Netherlands). X-ray photoelectron spectroscopy (XPS) (Waltham, MA, USA) was employed to analyze the elemental composition of the corrosion product.

## 3. Results

### 3.1. Corrosion Rate Text

Figure 2 depicts the temporal evolution of the carbon steel corrosion rate under various operating conditions.

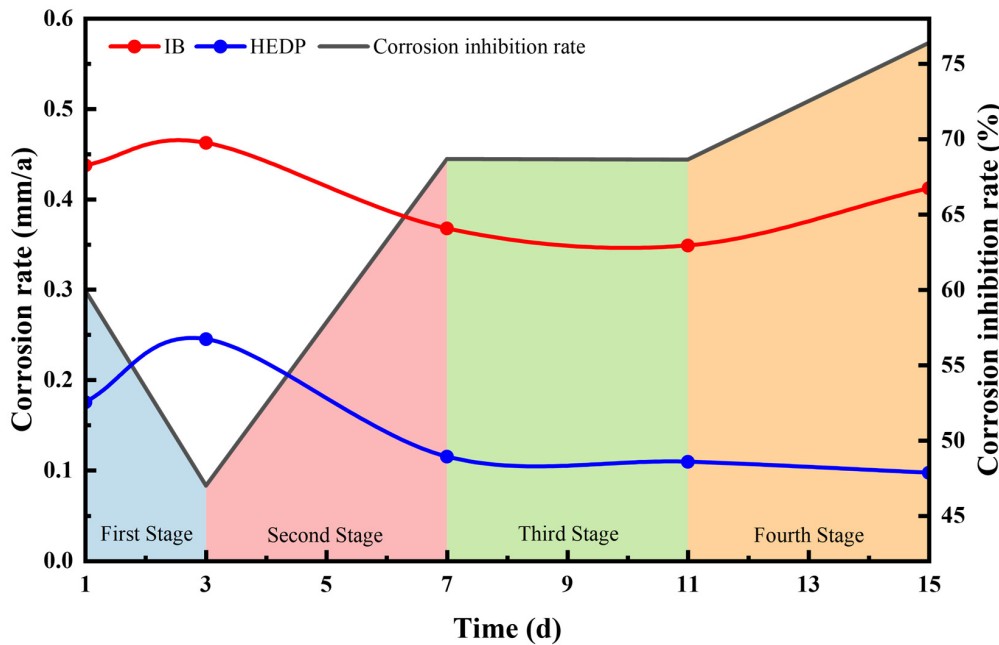

**Figure 2.** Corrosion rate of carbon steel.

According to Figure 2, the corrosion rate of carbon steel under the IB+HEDP condition was approximately 60% lower than that under the IB working condition, suggesting that HEDP effectively reduced the corrosion rate in the presence of iron bacteria, surpassing the corrosion inhibition effect observed in the SRB system [17]. The corrosion inhibition behavior of HEDP in the iron bacteria system reveals distinct phases.

First stage (1–3 days): Both operating conditions experienced an increase in corrosion rates. The IB condition saw a 2.86% rise, whereas the IB+HEDP condition exhibited a sharper increase of 20.94%. This indicates that HEDP initially accelerated the corrosion reaction rate of carbon steel in the presence of iron bacteria, thereby diminishing the corrosion inhibition effect.

Second stage (3–7 days): Corrosion rates decreased in both conditions, with the IB condition decreasing by 20.48% and the IB+HEDP condition by 53.02%. This suggests that during this stage, HEDP effectively mitigated the corrosion reaction rate of carbon steel in the iron bacteria system, enhancing the corrosion inhibition rate.

Third stage (7–11 days): Corrosion rate changes were minimal (less than 4%) in both conditions, indicating stability. HEDP did not significantly alter the corrosion reaction rate of carbon steel in the iron bacteria system, maintaining a steady corrosion inhibition rate.

Fourth stage (11–15 days): While the corrosion rate increased by 18.17% in the IB condition, it decreased by 11.05% in the IB+HEDP condition. This significant difference suggests that HEDP's presence inhibited carbon steel corrosion under the iron bacteria system during this stage, leading to a renewed increase in the corrosion inhibition rate.

In summary, the corrosion inhibition behavior of HEDP on carbon steel under the iron bacteria system follows a pattern of weakening, strengthening, stabilization, and then strengthening again. The presence of iron bacteria, along with certain factors, such as EPS content and composition, plays a crucial role in carbon steel corrosion [28]. To delve deeper into the mechanism of HEDP corrosion inhibition, further investigation into its impact on the number of iron bacteria, EPS content, and composition is warranted.

*3.2. IB Characteristics Analysis*

Figure 3 illustrates the variations in the number of suspended and attached iron bacteria over time across different operating conditions during the experiment. The graph highlights a notable observation: under the IB+HEDP condition, the average count of iron bacteria is one order of magnitude higher compared to the IB condition alone. This suggests that HEDP facilitates the growth of iron bacteria. Notably, this finding aligns with the conclusions drawn in the study conducted by Kimbell et al. [29]. Based on the growth rate of iron bacteria, it can be inferred that prior to the decline of iron bacteria, HEDP enhances the growth of suspended iron bacteria by approximately 20%–154% and the growth of attached iron bacteria by around 78%–348%.

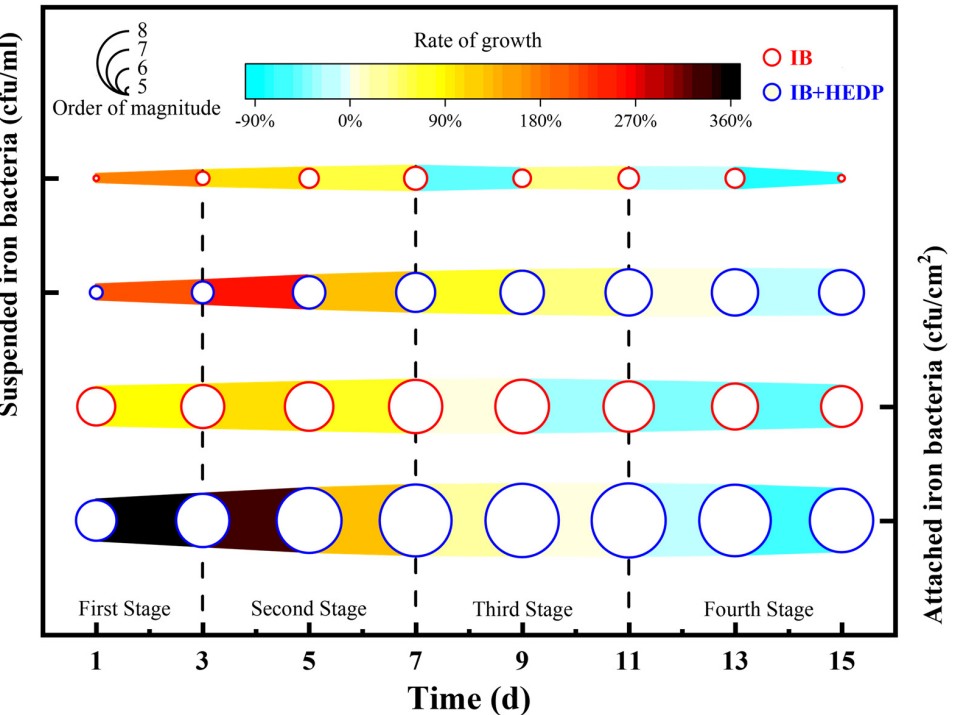

**Figure 3.** The number of iron bacteria.

According to Figure 3, during the first stage, the growth rate of suspended iron bacteria in the IB+HEDP condition is 1.2 times higher than that in the IB condition, while the growth rate of attached iron bacteria is 4.5 times higher, indicating that HEDP has a more pronounced promoting effect on iron bacteria at the carbon steel interface. Consequently, iron bacteria at the carbon steel interface exhibit higher activity and more vigorous metabolic activity. This observation aligns with the 18.08% increase in corrosion rate under

the IB+HEDP condition compared to the IB condition, as illustrated in Figure 2, suggesting that HEDP promotes corrosion by stimulating the metabolic activity of iron bacteria.

In the second stage, the growth rate of suspended iron bacteria in the IB+HEDP condition escalated from 1.2 times in the IB condition to 3.24 times. However, the growth rate of attached iron bacteria decreased from 4.5 times in the IB condition to 3.6 times. This indicates that the activity of attached iron bacteria in the IB+HEDP condition was inhibited during this stage. Previous studies have demonstrated that HEDP inhibits the metabolic processes of iron bacteria involved in electron transfer and reduces their activity within dense biofilms [30].

Therefore, the observed increase in the corrosion inhibition rate during this stage in Figure 2 may be attributed to the combined effect of a dense biofilm at the carbon steel interface and the reduced activity of attached iron bacteria.

In the third stage, under IB conditions, the number of iron bacteria decreased in both suspended and attached states, signaling the onset of a decay period in their growth and a significant decline in their activity. Additionally, on the 11th day, the number of suspended iron bacteria surpassed that on the 9th day, suggesting biofilm damage and cracking, leading to the detachment of attached iron bacteria from the carbon steel interface and their transformation into suspended iron bacteria [31].

Compared to conditions with only iron bacteria (IB condition), under conditions with both iron bacteria and HEDP (IB+HEDP condition), although the quantity of iron bacteria at the solution–carbon steel interface keeps rising, their growth rate diminishes. This suggests that HEDP solely sustains an increase in iron bacteria quantity during this phase without reversing the declining metabolic activity trend. Simultaneously, the growth rate of suspended iron bacteria unexpectedly surpasses that of attached iron bacteria by fivefold, indicating potential biofilm cracking and damage in the IB+HEDP operational state.

According to Figure 2, the corrosion promotion effect resulting from biofilm rupture is balanced by the corrosion inhibition effect formed by the weakened metabolic activity of iron bacteria. Consequently, the corrosion rates under both operating conditions enter a plateau period.

In the fourth stage, the number of iron bacteria in both operating conditions continued to decrease. In comparison with the IB condition, the decay rate of suspended iron bacteria in the IB+HEDP condition decreased by 64%, while the decay rate of attached iron bacteria increased by 3%. This suggests that in the IB+HEDP condition, the iron bacteria begin to deteriorate at this stage. In the solution, HEDP appears to stimulate the metabolic activity of iron bacteria, thereby reducing their decay rate. At the carbon steel interface, HEDP inhibits the activity of iron bacteria and accelerates their decay. According to the literature, a dense biofilm is a crucial factor leading to HEDP inhibiting the activity of iron bacteria [30]. It can be inferred that HEDP repaired the damaged biofilm in the previous stage, reformed a dense structure, and further inhibited the metabolic activity of iron bacteria. Consequently, the corrosion inhibition effect of HEDP in Figure 2 is the outcome of the combined effect of the repaired dense biofilm and the reduced metabolic activity of iron bacteria.

### 3.3. Biofilm EPS and EPS Component Analysis

EPS, a significant component of biofilms, is predominantly secreted by iron bacteria. Figure 4 illustrates the fluctuations in EPS, polysaccharides, and protein content within the biofilm over time under various operational conditions during the experiment. It is evident from the figure that under the IB+HEDP condition, the EPS content in the biofilm consistently surpasses that under the IB condition, suggesting that HEDP has the capacity to augment the overall EPS production within the biofilm.

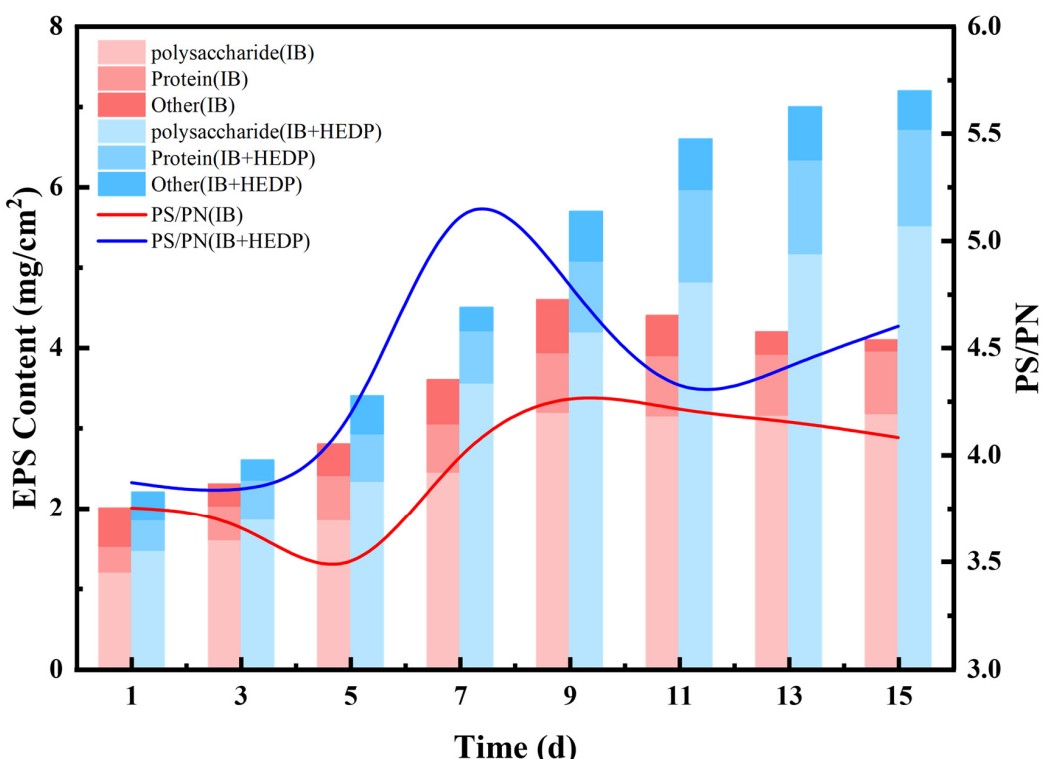

**Figure 4.** The content of EPS and its components in biofilms.

According to Figure 4, during the first stage, the content of EPS in the biofilm increased by 15% under the IB condition and by 18% under the IB+HEDP condition, which correlates with the increasing trend of attached iron bacteria shown in Figure 3. Additionally, the variation amplitude of PS/PN values between the two operating conditions is less than 2%, with the difference between them being less than 3%. This indicates that during this stage, the EPS content and composition secreted by iron bacteria in the two working conditions are largely similar, and HEDP has no discernible effect on EPS.

In the second stage, under IB+HEDP conditions, the EPS content within the biofilm surged by 73%. Research has shown that microorganisms can withstand harsh environmental conditions by secreting extracellular polymeric substances (EPS) [32]. Koju et al. suggest that HEDP demonstrates toxic effects on certain microorganisms, prompting them to secrete additional EPS to envelop themselves and avoid direct contact with the HEDP [33]. As outlined in Section 3.2, the toxic impact of HEDP diminishes the metabolic activity of certain attached iron bacteria during this stage, leading them to secrete more EPS into the biofilm to minimize contact with HEDP.

Simultaneously, the PS/PN value of the IB+HEDP condition in this stage increased from 3.80 to 5.48, marking a 34% rise compared to the maximum value observed under the IB condition in the same stage. This indicates that under HEDP stress, iron bacteria secrete EPS with greater viscosity, enhancing the stability and density of the biofilm [34].

In the third stage, under IB conditions, the EPS content in the biofilm peaked at 4.6 mg/cm$^2$ on the 9th day before gradually declining. As outlined in Section 3.2, this decrease can be attributed to the decay of iron bacteria, a reduction in their numbers, and a corresponding drop in EPS secretion. However, under IB+HEDP condition, HEDP sustains the growth of iron bacteria during this stage, preventing a decrease in their population. Consequently, we observed a 24% decrease in the PS/PN value under the IB+HEDP condition during this stage, suggesting that the biofilm rupture is also associated with a reduction in biofilm viscosity.

In the fourth stage, under the IB condition, the EPS content of the biofilm decreased by 10%, accompanied by a continued decrease in the PS/PN value. Through a comprehensive analysis of Sections 3.1 and 3.2, it can be deduced that iron bacteria during this phase

are undergoing decay, with a progressive decline in activity. Consequently, there is a weakening in their ability to secrete EPS and its viscous substances, leading to continuous biofilm damage and increased corrosion.

In contrast, under the IB+HEDP condition, the EPS content of the biofilm increased by 26%, with the PS/PN value also rising by 10%. This suggests an enhancement in the biofilm's adhesion ability and the potential for repair of damaged biofilms, thereby slowing down corrosion. These findings align with the experimental results presented in Sections 3.1 and 3.2.

### 3.4. EIS Analysis

Electrochemical impedance spectroscopy (EIS) technology stands out as one of the most suitable techniques for investigating the formation and growth of membranes. Figure 5 depicts the Nyquist plots under various operating conditions, with the fitted equivalent circuit represented as R (Q (R (QR))).

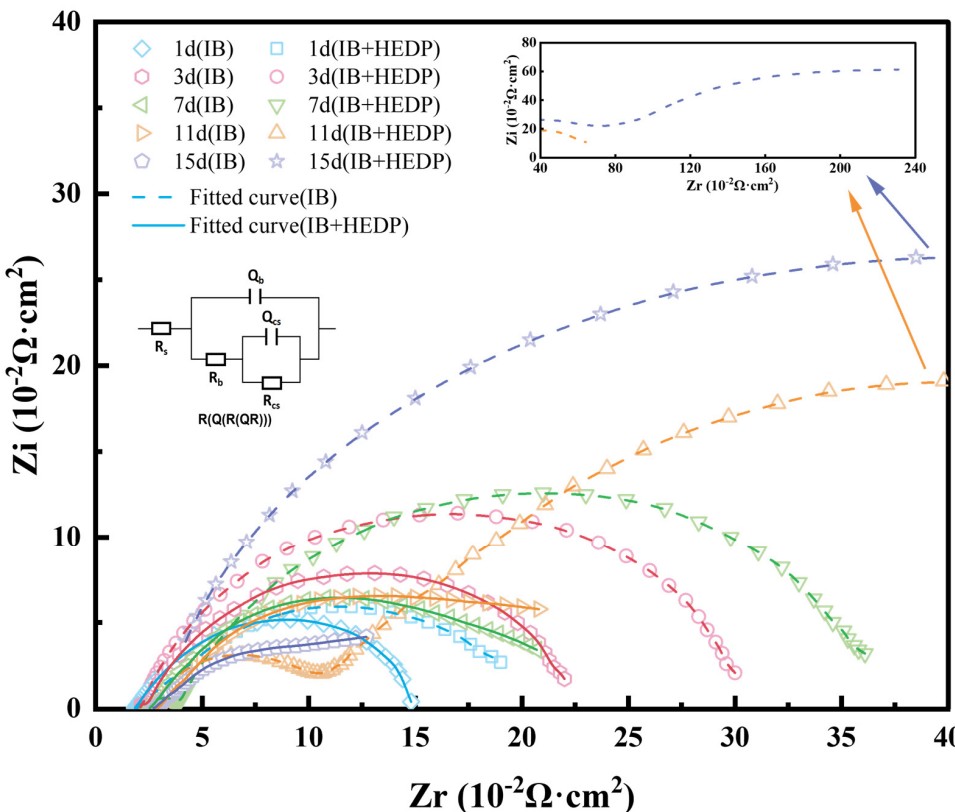

**Figure 5.** Nyquist plots and fitted equivalent circuit.

In the equivalent circuit, $R_s$, $R_b$, and $R_{cs}$ denote the solution resistance, the biofilm resistance, and the carbon steel interface resistance, respectively. Additionally, $Q_b$ and $Q_{cs}$ represent biofilm capacitance and carbon steel interface capacitance, respectively. The parameters $n_b$ and $n_{cs}$ stand for mass transfer coefficients, with values closer to 1 indicating better uniformity of the interface between the biofilm and carbon steel. Detailed fitting results are presented in Table 2.

**Table 2.** The electrochemical impedance parameters obtained by the fitted equivalent circuit.

| Time (d) | $R_s$ ($\Omega \cdot cm^2$) | $Q_b$ ($F \cdot cm^2$) | $n_b$ | $R_b$ ($\Omega \cdot cm^2$) | $Q_{cs}$ ($F \cdot cm^2$) | $n_{cs}$ | $R_{cs}$ ($\Omega \cdot cm^2$) |
|---|---|---|---|---|---|---|---|
| | | | | **IB** | | | |
| 1 | 185.2 | - | - | - | $5.2 \times 10^{-5}$ | 0.8315 | 2456 |
| 3 | 206.0 | $4.8 \times 10^{-5}$ | 0.8812 | 445 | $7.2 \times 10^{-3}$ | 0.8172 | 2035 |
| 7 | 258.6 | $5.1 \times 10^{-4}$ | 0.7472 | 2237 | $4.7 \times 10^{-3}$ | 0.9887 | 2857 |
| 11 | 293.2 | $8.7 \times 10^{-4}$ | 0.7091 | 2742 | $1.3 \times 10^{-4}$ | 1.0000 | 4462 |
| 15 | 274.5 | $1.5 \times 10^{-3}$ | 0.6238 | 1236 | $1.0 \times 10^{-3}$ | 1.0000 | 2924 |
| | | | | **IB+HEDP** | | | |
| 1 | 190.0 | - | - | - | $4.8 \times 10^{-4}$ | 0.7789 | 4032 |
| 3 | 188.2 | $2.4 \times 10^{-5}$ | 0.7754 | 1087 | $8.2 \times 10^{-4}$ | 0.8398 | 3687 |
| 7 | 257.4 | $3.1 \times 10^{-4}$ | 0.9405 | 3213 | $8.6 \times 10^{-5}$ | 0.8022 | 6751 |
| 11 | 287.6 | $1.9 \times 10^{-6}$ | 0.8737 | 4042 | $3.1 \times 10^{-6}$ | 0.6768 | 8230 |
| 15 | 304.1 | $1.4 \times 10^{-6}$ | 0.8000 | 8643 | $5.4 \times 10^{-7}$ | 0.8000 | 21,241 |

According to Table 2, the overlap rates of resistance values in different parts under the two working conditions vary. The coincidence rate of $R_s$ is 95%, indicating that HEDP has a minimal effect on the conductivity of circulating cooling water. However, the overlap rates of $R_b$ and $R_{cs}$ are 20% and 4%, respectively, with the values under the IB+HEDP condition being at least 43% higher than those under the IB condition (up to 626%). This suggests that HEDP enhances the difficulty of charge transfer at the interface between the biofilm and the carbon steel by altering the density of the biofilm and the activation energy of the carbon steel interface. This finding is consistent with the corrosion inhibition results of HEDP outlined in Section 3.1.

Furthermore, compared to the IB condition, the $n_b$ value of the IB+HEDP condition is larger, and the $n_{cs}$ value is smaller. This indicates that HEDP helps enhance the uniformity of the biofilm but exacerbates the non-uniformity of the carbon steel interface. This phenomenon is associated with the composition and configuration of corrosion products at the carbon steel interface, which will be further discussed in Sections 3.6 and 3.7.

According to Figure 5, after 7 days of the IB condition, the capacitance arc is no longer a complete semicircle, and the right side gradually leans towards a 45° diagonal line, indicating a decrease in impedance. This observation aligns with the experimental findings in Sections 3.2 and 3.3, suggesting that biofilm rupture is the primary reason for the impedance decrease.

Conversely, under the IB+HEDP condition, the capacitance arc remains intact, and the capacitance radius is large. This observation further supports the hypothesis in Section 3.2 that HEDP has a repairing effect on biofilms.

### 3.5. Potentiodynamic Polarization Curve Measurements

Table 3 presents the corrosion current density and the Tafel slopes derived from the polarization curves (Figures S1 and S2) on the carbon steel electrode under various operating conditions. It is evident that the magnitude and trend of the corrosion current in both working conditions are consistent with the corrosion rate depicted in Figure 2.

Figure 6 illustrates the variation of the Tafel slope (absolute value) over time under different operating conditions. Notably, irrespective of the operating conditions, the absolute value of the cathode slope consistently surpasses that of the anode, suggesting that the corrosion process of carbon steel under the iron bacteria system is primarily governed by the cathode [35–37]. Furthermore, the cathodic curve of the IB+HEDP condition exhibited a significantly higher slope than that of the IB condition throughout the experiment, indicating that cathodic inhibition is the primary corrosion inhibition mechanism of HEDP.

**Table 3.** Corrosion current and Tafel slopes in different conditions.

| Time (d) | $I_{corr}$ ($\mu A/cm^2$) | $\beta_a$ (mV/dec) | $-\beta_c$ (mV/dec) |
|---|---|---|---|
| **IB** | | | |
| 1 | 6.779 | 87 | 176 |
| 3 | 7.622 | 121 | 209 |
| 7 | 4.495 | 104 | 220 |
| 11 | 3.808 | 105 | 242 |
| 15 | 5.562 | 80 | 233 |
| **IB+HEDP** | | | |
| 1 | 2.321 | 99 | 326 |
| 3 | 3.401 | 119 | 376 |
| 7 | 1.936 | 113 | 447 |
| 11 | 1.352 | 144 | 342 |
| 15 | 0.892 | 325 | 354 |

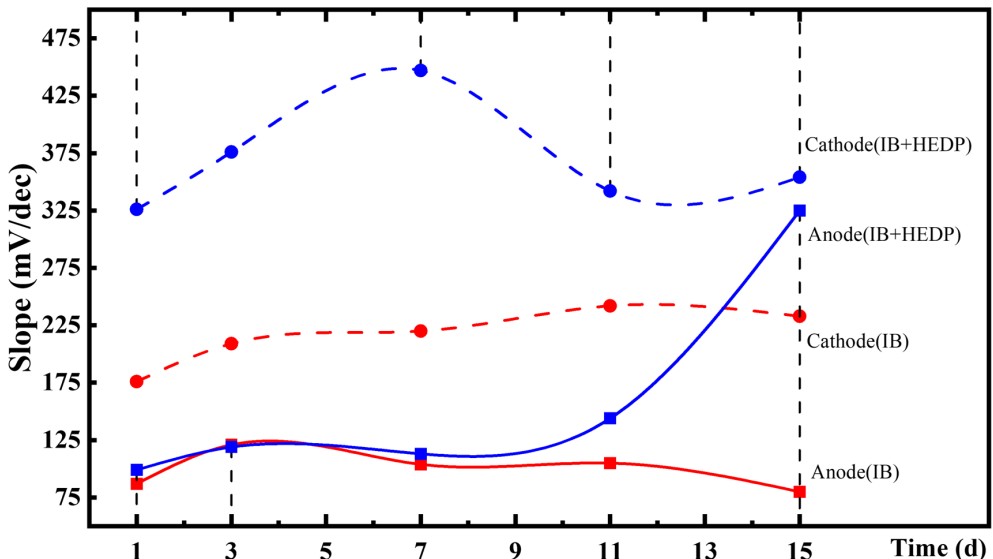

**Figure 6.** Slope of anode and cathode.

According to Figure 6, during the initial period from day 1 to day 7, the biofilm of both operating conditions remains intact. The addition of HEDP enhances the biofilm's oxygen-blocking capability by augmenting the EPS content and increasing the biofilm viscosity, consequently bolstering the cathodic inhibition effect. As a result, the cathodic slope in the IB+HEDP condition notably surpasses that in the IB condition, with the gap between the two widening gradually. This observation aligns with the Rb value test results outlined in Section 3.4.

Between days 7 and 11, both biofilms incur damage under both operating conditions. However, HEDP promptly repairs the biofilm, maintaining significant cathodic inhibition in the IB+HEDP condition. Nonetheless, during this phase, the disparity in the cathode slope between the two conditions gradually diminishes. As detailed in Section 3.2, the IB+HEDP condition hosts a significantly higher number of attached iron bacteria compared to the IB condition by more than one order of magnitude. Prior to the biofilm repair, the increased presence of iron bacteria accelerates the cathodic reaction rate, thus attenuating the cathodic inhibition effect.

From day 11 to 15, the biofilm structure is restored under the IB+HEDP condition, while the damage remains more severe in the IB condition. Consequently, the disparity in the cathode slope between the two conditions widens once again, augmenting the cathodic inhibition effect. This observation corroborates the Rb value test results outlined in Section 3.4.

In summary, the cathodic inhibition mechanism of HEDP primarily operates by impeding the transport of oxygen within the biofilm, thereby postponing the onset of corrosion. However, there exists a notable distinction in the change process between the anode and cathode slopes.

According to Figure 6, during the initial 1 to 3 days, the anode slopes for both operating conditions exhibit a similar trend, with the IB+HEDP condition slightly surpassing the IB condition. However, the discrepancy in slopes diminishes over time. Upon synthesizing the findings from Sections 3.2 and 3.4, it is apparent that while HEDP heightens the activation energy at the carbon steel interface, it also amplifies the activity of iron bacteria. The heightened activity of iron bacteria accelerates the rate of the anodic reaction, thereby weakening the effectiveness of anodic inhibition.

From days 3 to 15, the disparity between the anode slopes of the IB+HEDP and IB conditions gradually widens, indicating a pronounced enhancement in the anode suppression effect. By day 15, the difference between the anode slope and cathode slope of the IB+HEDP condition falls below 9%, suggesting that the anodic reaction becomes the predominant factor influencing the corrosion rate. This observation aligns with the abrupt rise in $R_{cs}$ values documented in Section 3.4, indicating an escalation in charge transfer resistance at the carbon steel interface. This phenomenon could potentially be linked to corrosion products and their chemical characteristics, a topic that will be further explored in Sections 3.6 and 3.7.

From this, it can be inferred that within the iron bacterial system, HEDP also demonstrates anodic inhibition. This inhibition is primarily achieved through the formation of various corrosion products at the carbon steel interface. Consequently, when electrons are lost at the interface, a higher activation energy is demanded, thus leading to a deceleration in the corrosion reaction rate.

*3.6. XPS Analysis*

In the corrosion products formed, both complexed HEDP and iron oxide contribute to increasing the activation energy of the carbon steel surface [38,39]. To ascertain the content of these two substances within the corrosion products, peak fitting analysis was conducted on the XPS full spectrum analysis of the corrosion products present on the carbon steel surface under various working conditions (Figures S3–S10). The fitting results for the principal elements are outlined in Table 4.

**Table 4.** Mass fraction of elements contained in corrosion products.

| Time (d) | Fe (%) | O (%) | C (%) | N/S/P (%) |
|:---:|:---:|:---:|:---:|:---:|
| **IB** | | | | |
| 3 | 22.71 | 46.22 | 30.97 | 0.10 |
| 7 | 25.22 | 45.93 | 28.85 | 0.00 |
| 11 | 29.62 | 45.40 | 26.25 | 0.12 |
| 15 | 27.14 | 45.78 | 26.80 | 0.28 |
| **IB+HEDP** | | | | |
| 3 | 13.21 | 41.25 | 39.78 | 5.76 |
| 7 | 15.48 | 39.83 | 44.56 | 0.13 |
| 11 | 17.26 | 44.15 | 35.93 | 2.66 |
| 15 | 20.32 | 41.33 | 35.16 | 3.19 |

EPS and HEDP within biofilms serve as the primary sources of N, P, and S elements found in corrosion products. From Table 4, it is evident that the proportion of N, P, and S elements in corrosion products under IB+HEDP conditions exceeds that under IB conditions by one order of magnitude. According to Section 3.3, the maximum ratio of EPS content in the biofilm during the experimental period was 1:1.7, which is significantly lower than one order of magnitude. Consequently, the disparity in the proportion of N, P, and S elements in

the corrosion products between the two working conditions can be utilized to characterize changes in HEDP content within the corrosion products.

The types and proportions of iron oxides present on the carbon steel surface can be directly inferred from the Fe2p peak of the Fe element. The results of the two peak fitting analyses were collated, and the variations in the proportion of HEDP and various iron oxides within corrosion products over time under different operating conditions are depicted in Figure 7.

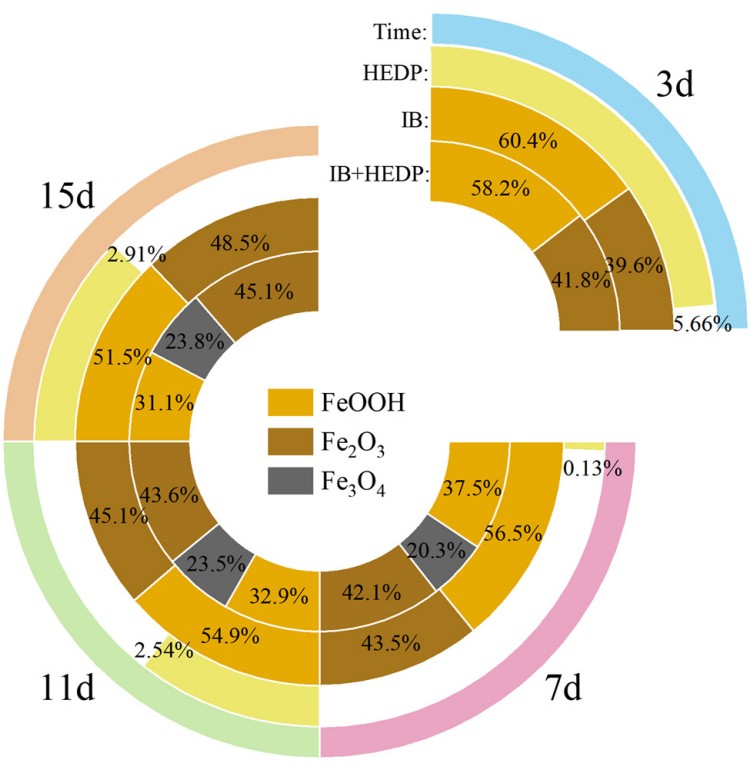

**Figure 7.** The content of HEDP and the proportion of various iron oxides in corrosion products.

According to Figure 7, at the 3-day mark, the proportion of HEDP in the biofilm under IB+HEDP conditions peaked at 5.66%, representing the highest accumulation observed during the experiment's duration. In the first stage, the utilization of HEDP by iron bacteria is minimal, allowing for greater chelation of HEDP within the biofilm. The iron oxides formed by corrosion in both working conditions during this phase consist primarily of FeOOH and $Fe_2O_3$, with both containing only $Fe^{3+}$. The iron bacteria system predominantly comprises iron-oxidizing bacteria (IOB) and iron-reducing bacteria (IRB) [25]. IOB facilitates the oxidation of $Fe^{2+}$ to $Fe^{3+}$, while IRB catalyzes the reduction of $Fe^{3+}$ to $Fe^{2+}$ in a microaerobic environment, with both processes promoting metal corrosion [40,41]. Consequently, IOB plays a key role in the formation of iron oxides during this corrosion stage. Based on the comprehensive analysis of Sections 3.1 and 3.2, it can be concluded that the primary reason for the decrease in the corrosion inhibition rate during the first stage is the utilization of a small amount of HEDP by IOB to enhance their activity within the IB+HEDP condition.

At 7 days, the proportion of HEDP in the biofilm under IB+HEDP conditions decreased significantly from 5.66% to 0.13%, representing the lowest value observed during the experiment. Initially, there is a high utilization of HEDP by iron bacteria, leading to its decomposition within the biofilm. Research by Edward M. Fox et al. suggests that the C–P bond of HEDP is relatively stable and not easily degraded by various chemical and biological processes, except under conditions where inorganic phosphorus is limited, thus allowing certain bacteria to effectively utilize it as a phosphorus source [42].

The dense biofilm formation in this stage restricts substance transport, thus creating a phosphorus- and oxygen-deficient environment and compelling attached iron bacteria to utilize HEDP as a phosphorus source. This hypoxic environment favors the metabolic growth of IRB, resulting in the appearance of $Fe_3O_4$ containing $Fe^{2+}$ in the iron oxide formed by corrosion under IB conditions. However, under IB+HEDP conditions, the iron oxides formed by corrosion are predominantly FeOOH and $Fe_2O_3$, indicating an inconsistency with the dense biofilm phenomenon.

Observations from Figure 7 reveal that approximately 35% of FeOOH did not convert to $Fe_3O_4$ as expected in the IB+HEDP condition, leading to an increase in FeOOH converted to $Fe_2O_3$. Li et al.'s study suggests that HEDP blocks the production of reducing bacterial iron transporters and reduces iron uptake [17]. Therefore, the comprehensive analysis of Sections 3.2 and 3.3 suggests that the utilization of HEDP by attached iron bacteria hampers the ability of IRB to obtain energy by reducing $Fe^{3+}$, consequently reducing their activity [43].

Moreover, the micro-anaerobic environment fostered by the dense biofilm is not conducive to the growth of IOB, further inhibiting their activity. Consequently, the activity of attached iron bacteria decreases, and the corrosion reaction rate slows down at this stage.

At 11 days, the proportion of HEDP in the biofilm under IB+HEDP conditions increased significantly from 0.13% to 2.54%, nearly reaching 50% of its peak. This suggests that during the third stage, the utilization of HEDP by iron bacteria decreases, allowing some HEDP molecules to once again chelate with the biofilms. Through a comprehensive analysis of Sections 3.2 and 3.4, it can be concluded that the disrupted biofilm exhibits a diminished capacity to impede oxygen and inorganic phosphorus in the solution, thereby alleviating phosphorus deficiency within the biofilm and reducing the utilization of HEDP by attached iron bacteria. Simultaneously, oxygen replaces HEDP to inhibit the activity of iron-reducing bacteria (IRB), consequently preventing the formation of $Fe_3O_4$ under IB+HEDP conditions until the chelated HEDP fully restores the biofilm structure.

At 15 days, the proportion of HEDP in the biofilm under the IB+HEDP condition slightly increased from 2.54% to 2.91%, indicating that in the fourth stage, the utilization of HEDP by iron bacteria was improved, and only a small portion of HEDP continued to chelate with the biofilm. Based on the comprehensive analysis of Sections 3.3 and 3.4, it can be concluded that HEDP has repaired the damage to the biofilm and restored the phosphorus- and nutrient-deficient environment within the membrane. IRB uses HEDP as a phosphorus source again, secretes viscous EPS, and is unable to generate $Fe_3O_4$.

The alternation in inhibition caused by HEDP and oxygen impedes the process wherein IRB converts FeOOH to $Fe_3O_4$, leading to the absence of $Fe_3O_4$ formation at the carbon steel interface. However, $Fe_3O_4$, known for its dense structure and strong chemical stability, is less likely to lose electrons compared to FeOOH [44]. This observation contradicts the changes depicted in the anode curve illustrated in Figure 7.

In Zhang et al.'s research, it was noted that HEDP influenced the deposition and crystallization forms of surface crystals on carbon steel [45]. Hence, it can be inferred that the configurations of FeOOH and $Fe_2O_3$ formed on the carbon steel surface under the IB+HEDP condition might differ from those under the IB condition, thus contributing to the variations observed in the electrochemical behavior depicted in Figure 7.

### 3.7. SEM Analysis

Based on the analysis in Sections 3.4 and 3.5, a 15-day corroded carbon steel specimen was selected for SEM analysis of its surface. SEM electron microscopy images of iron oxides under different operating conditions are shown in Figure 8.

From Figure 8, it is evident that in the IB condition, the iron oxides predominantly exhibit a flake-like $\gamma$-FeOOH, alongside spherical $\gamma$-$Fe_2O_3$ and $\gamma$-$Fe_3O_4$ [46–48]. Conversely, in the IB+HEDP condition, the iron oxides primarily manifest as flocculent $\alpha$-$Fe_2O_3$ and honeycomb-like $\alpha$-FeOOH [49,50]. This indicates the formation of different structural iron oxides on the carbon steel surface under these different conditions.

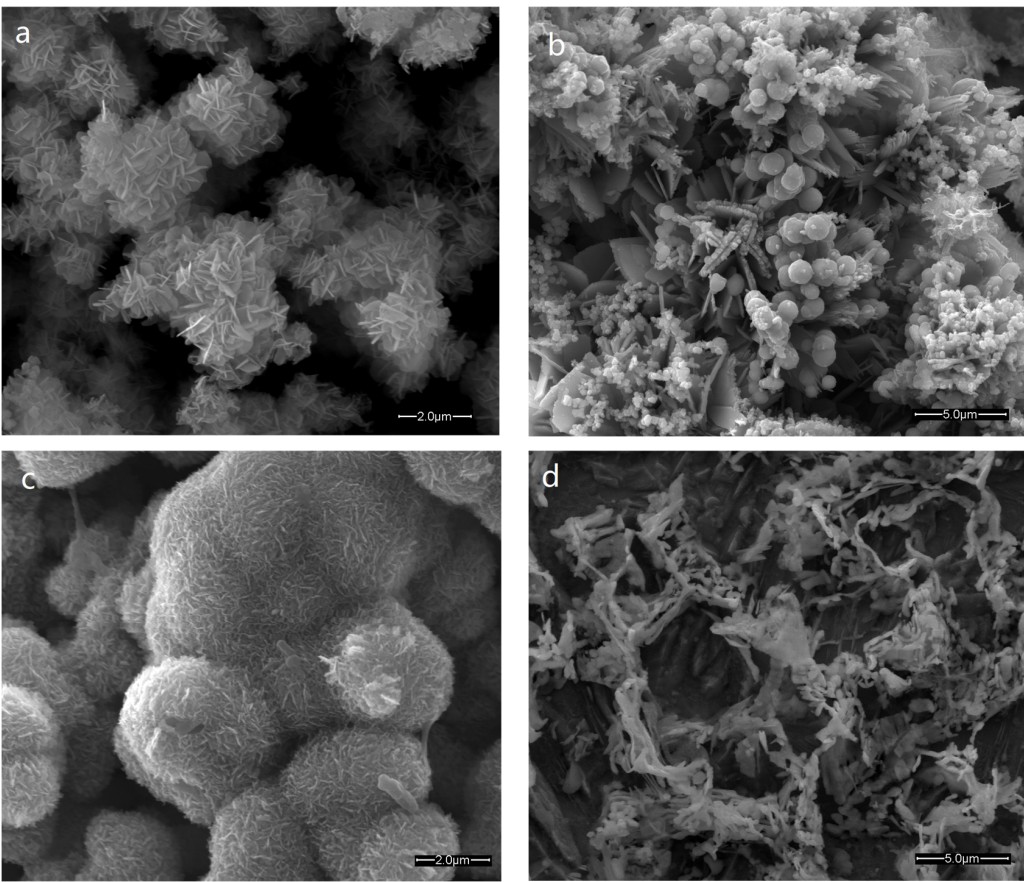

**Figure 8.** SEM images on 15 d: IB condition (**a**,**b**); IB+HEDP condition (**c**,**d**).

In line with Xiong et al.'s findings, the type of iron oxide formed during carbon steel corrosion varies depending on the ion concentrations present in the environment [51]. Specifically, when the concentration of $Fe^{2+}$ is high, $\gamma$-type iron oxide is preferentially formed. On the other hand, when the concentration of $Fe^{3+}$ is elevated, $\alpha$-type iron oxide is favored. This distinction underscores the role of environmental ion concentrations in influencing the composition and structure of iron oxides formed during carbon steel corrosion, aligning with the observations made in Figure 8.

During the experiment, $Fe^{3+}$ uptake by IRB was continuously inhibited, while IOB converted $Fe^{2+}$ into $Fe^{3+}$, thus gradually increasing $Fe^{3+}$ concentration on the carbon steel surface and fostering an environment for $\alpha$-shaped iron oxide formation. After biofilm rupture, oxygen entry boosted IOB activity, accelerating $Fe^{3+}$ stacking and $\alpha$-shaped iron oxide formation. Studies confirm that $\alpha$-shaped iron corrosion products exhibit higher chemical stability than $\gamma$-shaped ones [52], which is consistent with the results of Sections 3.4 and 3.5. Ultimately, $\alpha$-shaped iron corrosion products coat the steel surface, inducing anodic inhibition.

## 4. Discussion

HEDP inhibits the activity of iron bacteria, alters the structure of iron bacterial biofilm, and affects the charge transfer at the carbon steel interface, resulting in a change in the corrosion inhibition effect of HEDP on carbon steel under the iron bacterial system over time.

In the first stage, XPS analysis revealed a gradual accumulation of HEDP within the biofilm, enhancing its density as the biofilm developed. During this phase, a corrosion inhibition mechanism primarily centered on cathodic inhibition emerged, resulting in an overall reduction in the corrosion rate. This phenomenon was further confirmed by EIS testing, which indicated a notable increase in $R_b$ compared to $R_{cs}$. However, despite the corrosion inhibition effect, HEDP facilitated the proliferation of iron bacteria, leading to a



substantial rise in the number of attached iron bacteria. During this stage, characterized by an increase in corrosion rate, the abundant presence of attached iron bacteria further accelerates the corrosion reaction [53]. Consequently, the corrosion rate under the IB+HEDP condition increases at a faster pace.

In the second stage, HEDP contributes to enhancing the biofilm density, thereby impeding the transport of inorganic phosphorus and oxygen. Iron bacteria attached to surfaces start utilizing HEDP as a phosphorus source. Nevertheless, the micro-anaerobic environment created inhibits the activity of IOB, and the utilization of HEDP as a phosphorus source obstructs IRB from acquiring $Fe^{3+}$ for energy metabolism, consequently restraining IRB activity. This stage marks a period of decreased corrosion rate. The subdued activity of attached iron bacteria decelerates the corrosion reaction, resulting in a more rapid decline in the corrosion rate under the IB+HEDP condition and an augmentation in the corrosion inhibition rate.

XPS analysis results further validate this phenomenon, indicating a near depletion of HEDP in the biofilm, without the generation of $Fe_3O_4$. The reduction of HEDP within biofilms may lead to a decrease in membrane density. However, under stress, IRB secretes additional viscous EPS to uphold biofilm density. EIS test outcomes demonstrate an improved biofilm density to some extent, thereby highlighting cathodic inhibition as the primary corrosion inhibition mechanism during this stage.

In the third stage, the transient surge in suspended iron bacteria and the notable reversal of their growth rate in both operational settings suggest that the biofilm structure undergoes hydraulic damage [54]. Consequently, a substantial portion of iron bacteria detaches from the biofilm, transitioning from an attached to a suspended state. Simultaneously, within the solution, inorganic phosphorus and oxygen seize the opportunity to infiltrate the biofilm, displacing HEDP and hindering the activity of IRB. This substitution not only diminishes the secretion of viscous EPS but also amplifies the activity of IOB, thereby attenuating the cathodic inhibition effect.

However, XPS analysis reveals that HEDP re-enters the biofilm and accumulates anew. EIS test results indicate that HEDP ultimately reinstates the biofilm's density. The inhibition of IRB activity and the enhancement of IOB activity lead to an elevation in $Fe^{3+}$ concentration on the carbon steel surface. SEM images depict the gradual formation of a more stable $Fe^{3+}$ concentration $\alpha$-type iron oxide layer on the carbon steel surface, augmenting the anodic suppression.

Consequently, under the IB+HEDP condition, cathodic suppression weakens, while anodic suppression strengthens, resulting in an unchanged overall corrosion rate. In the IB condition, biofilm rupture exacerbates corrosion. However, the decay of attached iron bacteria significantly offsets this effect, yielding a stable cathode slope and overall corrosion rate. The corrosion inhibition rate remains consistent at this stage compared to the two operational conditions.

In the fourth stage, all iron bacteria have entered a state of decay. With HEDP repairing the damaged biofilm, a micro-anaerobic environment re-emerges, inhibiting IOB activity. IRB also resumes using HEDP as a phosphorus source, further hindering its activity.

This accelerates the decay of attached iron bacteria and reduces the decomposition of HEDP. XPS analysis reveals further HEDP accumulation within the biofilm, which, combined with viscous EPS from IRB, enhances biofilm density, resulting in increased cathodic inhibition. EIS testing indicates the gradual coverage of the carbon steel surface by $\alpha$-shaped iron oxide, significantly impeding charge transfer at the interface. Polarization curve fitting results highlight the anode's increasing role in corrosion control at this stage.

In the IB+HEDP condition, the increased biofilm density and hindered electron loss at the carbon steel interface lead to a further decrease in the corrosion rate. Conversely, in the IB condition, the loss of biofilm protection results in a gradual increase in the corrosion rate [55]. Ultimately, compared to both conditions, the corrosion inhibition rate increases in this stage.

In summary, the corrosion inhibition behavior and mechanism of HEDP under the iron bacteria system are shown in Figure 9.

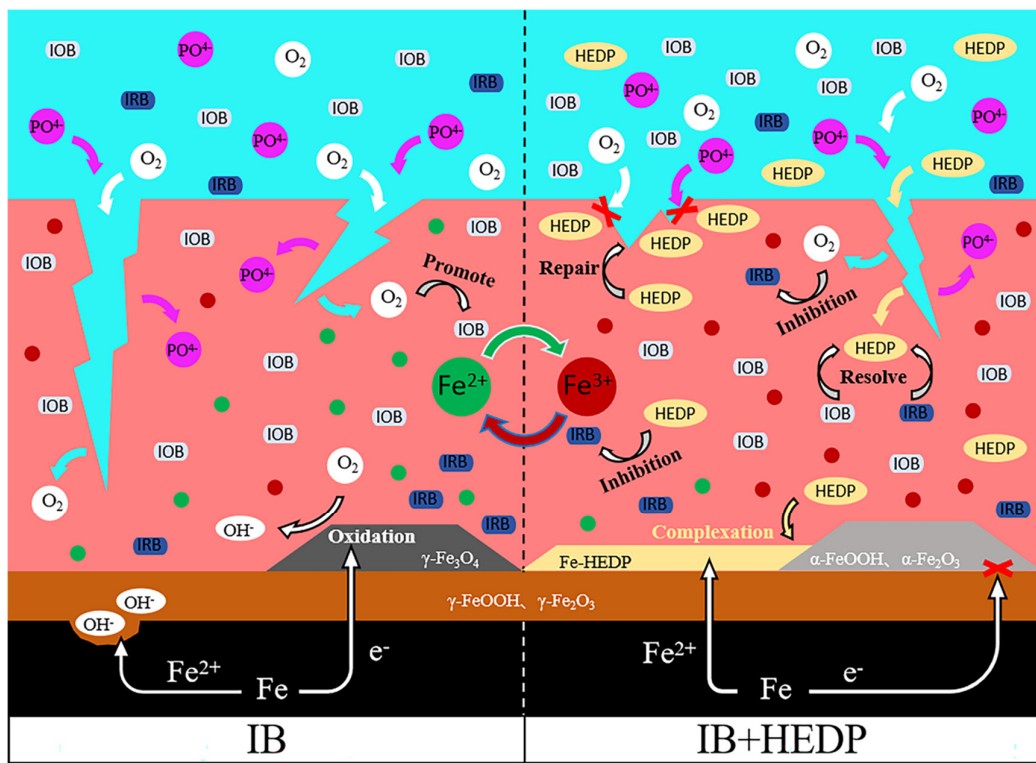

**Figure 9.** The corrosion inhibition mechanism of HEDP in an iron bacteria system.

## 5. Conclusions

In the iron bacteria system, HEDP demonstrates an average corrosion inhibition rate of 60% from day 1 to day 15, effectively retarding carbon steel corrosion. However, HEDP's influence on the two primary iron bacteria, IOB and IRB, yields inconsistent effects, thereby causing the corrosion inhibition efficacy of HEDP to fluctuate throughout the experiment:

◆ First stage (1–3 days): Oxygen inhibits the activity of IRB, while HEDP promotes the activity of IOB, resulting in a decrease in the corrosion inhibition rate.

◆ Second stage (3–7 days): HEDP inhibits the activity of IRB and stimulates the additional secretion of viscous EPS by IRB, thus enhancing the density of the biofilm, inhibiting the activity of IOB, and increasing the corrosion inhibition rate.

◆ Third stage (7–11 days): Oxygen once again inhibits the activity of IRB, while HEDP continues to promote the activity of IOB. However, IRB still does not reduce $Fe^{3+}$ in the environment. Gradually, the carbon steel interface transitions to predominantly chemically stable $\alpha$-iron oxide, maintaining a stable corrosion inhibition rate.

◆ Fourth stage (11–15 days): HEDP once again inhibits the activity of IRB and further suppresses the activity of IOB. Meanwhile, the carbon steel interface is already covered by $\alpha$-iron oxide. This results in another increase in the corrosion inhibition rate.

Ultimately, after undergoing a process of weakening, strengthening, stabilization, and further enhancement, the corrosion inhibition mechanism of HEDP is established, with cathodic inhibition as the primary mechanism and anodic inhibition as the secondary mechanism. The highest corrosion inhibition rate reaches 76%, showing a significant improvement compared to the SRB system.

**Supplementary Materials:** The following supporting information can be downloaded at: https://www.mdpi.com/article/10.3390/coatings14050580/s1.

**Author Contributions:** Conceptualization, P.X.; Methodology, Y.Z. and P.B.; Software, Y.Z.; Validation, P.X.; Formal Analysis, Y.Z. and P.B.; Investigation, Y.Z. and P.B.; Resources, P.X.; Data Curation, Y.Z. and P.B.; Writing—Original Draft Preparation, Y.Z.; Writing—Review and Editing, P.X.; Visualization, Y.Z. and P.B.; Supervision, P.X.; Project Administration, P.X.; Funding Acquisition, P.X. All authors have read and agreed to the published version of the manuscript.

**Funding:** This research was funded by the National Natural Science Foundation of China, grant number No. 51578035, and the National Major Water Pollution and Treatment Project of China, grant number No. 2018ZX07110-008-006.

**Institutional Review Board Statement:** Not applicable.

**Informed Consent Statement:** Not applicable.

**Data Availability Statement:** The data presented in this study are available on request from the corresponding author. The data are not publicly available due to privacy.

**Acknowledgments:** Thanks to the bacterial repository of Huazhong University of Science and Technology for providing the original bacterial strains.

**Conflicts of Interest:** The authors declare no conflicts of interest.

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
