# Peer review of "Research on the Corrosion Inhibition Behavior and Mechanism of 1-Hydroxy-1,1-ethyledine Disodium Phosphonate under an Iron Bacteria System"

_coatings, doi:10.3390/coatings14050580_

Round 1
Reviewer 1 Report
Comments and Suggestions for Authors
This article reports a complete investigation of the utilization of 1-hydroxyethylidene-1,1-diphosphoric acid (HEDP) to inhibit the corrosion of carbon steel caused by bacteria. The topic of this manuscript is interesting and attracts a broad readership in the field of polymer coatings and applications. Overall, this work is well written. The strengths of this review are as follows:
- Methodologies are comprehensive.
- The characterization results are complete as the samples are identified by various techniques, e.g., electrochemical, spectroscopic, and microscopic studies. Therefore, the authors can provide insight and accurate mechanisms, as mentioned in the title and discussion sections.
- The conclusion is concise and precise.
However, there are three issues that are required to be improved prior to publication.
(1) The introduction is not very well organized. The authors are suggested to rewrite the introduction section.
(2) Please check the y-axis label of Figure 6 (Slope of Slope) to see whether it is correct or not.
(3) Please also give the Tafel slope finding figures in the supporting information file.
Author Response
Thank you very much for taking the time to review this manuscript amidst your busy schedule. Every comment you have provided is very valuable. We have further improved the manuscript based on your feedback and suggestions. The following are detailed responses to each comment, and the parts that have been improved are highlighted in the resubmitted manuscript in red.

Reviewer 2 Report
Comments and Suggestions for Authors
The following issues must be discussed:
1. On the last part of the Introduction the authors must clearly state what is new or innovative in this work compared with other similar studies;
2. In what extend HEDP facilitates the growth of iron bacteria?;
3. Is not clear how HEDP delays the decay of iron bacteria but in the same time it is not able to halt the trend of decreased metabolic activity;
4. Explain in more details how HEDP has the capacity to augment the overall EPS production within the biofilm;
Author Response

(The authors gave the same response as above.)

Round 2
Reviewer 2 Report
Comments and Suggestions for Authors
The manuscript can be published in present form.